# Competing-risks model for predicting the prognostic value of lymph nodes in medullary thyroid carcinoma

**Fangjian Shang** [1]☯, **Xiaodan Liu**[1]☯, **Xin Ren**[1], **Yanlin Li**[1], **Lei Cai**[1], **Yujia Sun**[1], **Jian Wen**[1‡], **Xiaodan Zhai**[2‡]*

1 Department of General Surgery, The Fourth Affiliated Hospital of China Medical University, Shenyang, Liaoning Province, China, 2 Department of Endocrine, Shengjing Hospital of China Medical University, Shenyang, Liaoning Province, China

☯ These authors contributed equally to this work.
‡ These authors also contributed equally to this work
* zhaixiaodan0712@126.com

**Data Availability Statement:** The Surveillance, Epidemiology and End Results (SEER) program of the National Cancer Institute (NCI) is one of the largest publicly available and authoritative sources of data on cancer incidence and survival. This

## Abstract

### Background

Medullary thyroid carcinoma (MTC) is an infrequent form malignant tumor with a poor prognosis. Because of the influence of competitive risk, there may suffer from bias in the analysis of prognostic factors of MTC.

### Methods

By extracting the data of patients diagnosed with MTC registered in the Surveillance, Epidemiology, and End Results (SEER) database from 1998 to 2016, we established the Cox proportional-hazards and competing-risks model to retrospectively analyze the impact of related factors on lymph nodes statistically.

### Results

A total of 2,435 patients were included in the analysis, of which 198 died of MTC. The results of the multifactor competing-risk model showed that the number of total lymph nodes (19–89), positive lymph nodes (1–10,11–75) and positive lymph node ratio (25%-53%,>54%), age (46–60,>61), chemotherapy, mode of radiotherapy (others), tumor size (2-4cm,>4cm), number of lesions greater than 1 were poor prognostic factors for MTC. For the number of total lymph nodes, unlike the multivariate Cox proportional-hazards model results, we found that it became an independent risk factor after excluding competitive risk factors. Competitive risk factors have little effect on the number of positive lymph nodes. For the proportion of positive lymph nodes, we found that after excluding competitive risk factors, the Cox proportional-hazards model overestimates its impact on prognosis. The competitive risk model is often more accurate in analyzing the effects of prognostic factors.

study used SEER * stat 8.3.9 software to retrieve the follow-up data of patients with MTC from 1998 to 2016.

**Funding:** The author(s) received no specific funding for this work.

**Competing interests:** The authors have declared that no competing interests exist.

## Conclusions

After excluding the competitive risk, the number of lymph nodes, the number of positive and the positive proportion are the poor prognostic factors of medullary thyroid cancer, which can help clinicians more accurately evaluate the prognosis of patients with medullary thyroid cancer and provide a reference for treatment decision-making.

## 1 Introduction

In 2020, the global cancer data showed that thyroid cancer accounted for 3% of the total cancer incidence, accounting for the ninth of all cancer incidence rates, and the mortality rate was relatively low, accounting for 0.4% of all cancer deaths [1]. Medullary thyroid carcinoma (MTC) is a malignant tumor originating from parathyroid cells (C cells). It is related to the level of serum calcitonin, accounting for only 1% - 2% of thyroid cancer, but its mortality accounts for 14% of thyroid cancer, which is a pathological type of thyroid cancer with a poor prognosis [2–4].

From 1983 to 2012, the age-adjusted incidence of MTC increased significantly from 0.14 cases per 100 thousand to 0.21 cases (P <0.001) [5]. Compared with differentiated thyroid carcinoma (DTC), MTC is more prone to lymph node (LN) metastasis after surgery [6]. Whether MTC has LN metastasis and the proportion of positive lymph nodes have been considered important prognostic factors [7–9]. As one of the prognostic factors, the effect of LN status on the prognosis of medullary thyroid carcinoma remains to be further studied after excluding the risk of competition.

Kaplan-Meier marginal regression and the Cox proportional-hazards model are widely used to identify prognostic risk factors in patients diagnosed with thyroid cancer [10,11]. However, cancer is not the only cause of death for cancer patients. There are many competitive events of non-cancer death (such as cardiogenic death, suicide, cerebrovascular accident, etc.), which are more evident in elderly patients [12–15]. According to our calculation, the competitive events of MTC cases in the SEER database accounted for 55.9% (251 / 449); the Cox proportional-hazards model will overestimate the incidence of outcome events. In this case, the competitive risk model can more accurately evaluate the relationship between predictive variables and outcome events [16,17].

Exploring the prognostic value of total lymph nodes (LNs), positive lymph nodes (PLNs), and the positive lymph node ratio (LNR) in MTC patients undergoing total thyroidectomy only within a single medical center could lead to higher selection bias than population-based data. Accordingly, this study aims to mitigate selection bias by utilizing SEER data to explore LNs, PLNs, and LNR's prognostic value through the competitive risk model in patients receiving total thyroidectomy for MTC.

We present the following article in accordance with the STROBE reporting checklist.

## 2 Materials and methods

### 2.1 Date collection and patient selection

The Surveillance, Epidemiology and End Results (SEER) program of the National Cancer Institute (NCI) is one of the largest publicly available and authoritative sources of data on cancer incidence and survival (Website address: https://seer.cancer.gov/) [18]. The study obtained post-treatment follow-up data for MTC patients from 1998 to 2016, through the use of SEER * stat 8.3.9 software. Everyone can access this data in the same manner as the authors, and the

authors did not have any special access privileges that others would not have. The data is public and does not involve patient privacy, so it does not need the review and consent of the ethics committee. The diagnosis of medullary thyroid carcinoma is based on the international classification of oncological diseases. Inclusion criteria were (1) the tumor site was thyroid, (2) the pathological classification was MTC (ICD histological Code: medullary carcinoma amloid stroma, medullary carcinoma NOS), and (3) the surgical method was total thyroidectomy. The exclusion criteria included: (1) survival time was 0, and (2) regional LNs were unavailable. (Fig 1).

The number of LNs (by interquartile classification: 0~2 vs 3~18 vs 19~89), PLNs (by interquartile classification: 0 vs 1~10 vs 11~75), LNR (by interquartile classification: 0~25% vs 25% ~53% vs 54%~), age (≤45 vs 46~60 vs ≥61), gender, race, marital status, diagnosis, AJCC stage (I~III vs IV vs Unknow), AJCC T (T0~T2 vs T3~T4 vs Tx/Unknow), AJCC N (N0 vs N1 vs Nx/Unknow), AJCC M stage (YES vs NO vs Unknow), chemotherapy, radiotherapy, radiotherapy method, pathological type, pathological grade, multiple lesions, history of previous malignant tumors, tumor size, survival time, survival outcome (OS / DSS and competitive risk outcome). The main variables are classified by the "survMisc" package on R software, and the

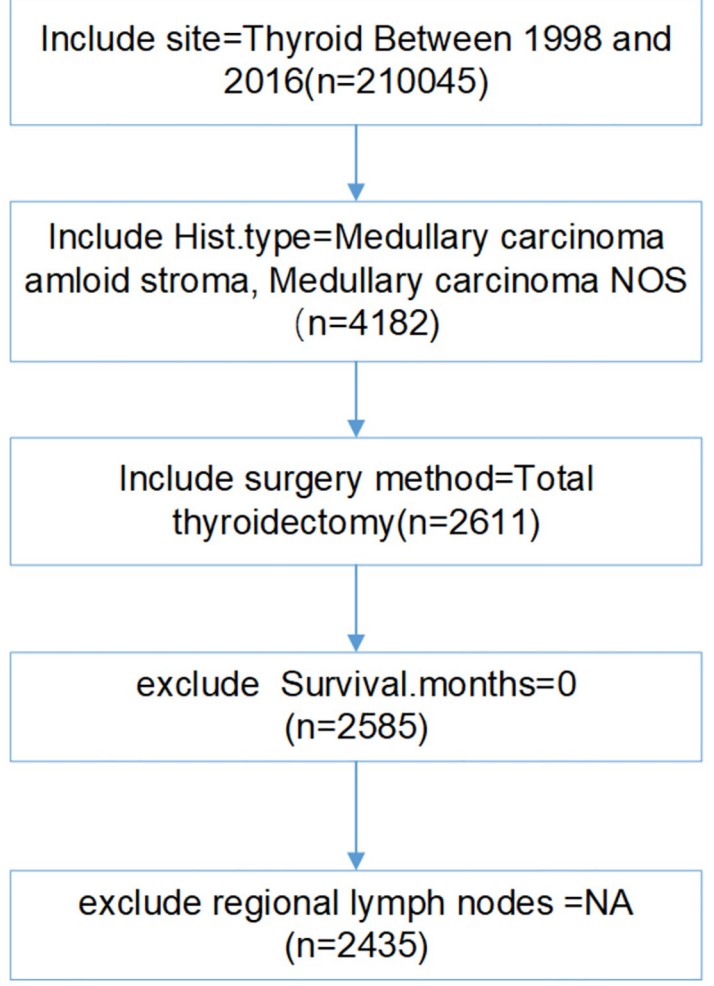

**Fig 1. Inclusion and exclusion criteria of the project.** Abbreviations: NOS, nothing of special; NA, Not available.

selection and classification of other variables are based on the guidelines and the results of other relevant studies [2,19,20].

The study evaluated the disease-free survival (DSS) and overall survival (OS) as the primary endpoints. We determined the specific cause based on the "SEER specific cause death classification" code in the SEER database. DSS refers to the interval between the date of diagnosis and the date of death only due to death or recent follow-up of MTC. OS is calculated from the date of diagnosis to the date of death caused by any cause or recent follow-up.

## 2.2 statistical analysis

This is a retrospective study that used OS and DSS as outcome indicators, the Kaplan–Meier method and log-rank test for survival analysis, and OS and DSS as outcome indicators for the univariate /multivariate Cox proportional-hazards model. We performed statistical analysis using the "survival" package in the R software and utilized the "forestplot" package to draw the forest plot. In the competitive risk analysis, we consider cancer-specific death and other causes of death as two competing events. We use the Fine-Gray model to conduct multivariate analysis to identify independent risk factors affecting the survival rate of medullary thyroid carcinoma and to establish clinical prediction models and risk scores. The "cmprsk" package on R software is used for analysis, and the "forest plot" package is used to draw the forest plot [21,22]. By comparing the results of the Cox proportional-hazards model and the Fine-Gray model, we can compare the impact of competitive risk factors on the survival rate of MTC and determine the more accurate prognostic factors for MTC. All statistical analyses were performed using R software version 4.1.1 (R Project, Vienna, Austria). All statistical tests were two-sided, with $P < 0.05$ indicative of statistical significance. Based on the SEER database and Cox proportional-hazards model analysis, the LNs, PLNs, and LNR in patients with MTC undergoing total thyroidectomy were included in the clinical prediction model. And through the establishment of the competing-risks model, eliminate the impact of competitive risk.

# 3 Results

## 3.1 Patient characteristics

A total of 2,435 patients participated in our study, with a median follow-up of 67 months. Among them, 198 died from MTC, while 251 deaths resulted from competing events, accounting for 55.9% of all deaths. Most patients were white (85.3%) and married (61.56%). There were significant differences in age, gender, diagnosis, AJCC stage, AJCC T, AJCC N, and AJCC M stage, chemotherapy, radiotherapy, radiotherapy mode, pathological type, pathological grade, multiple lesions, history of previous malignant tumors, tumor size, survival time, survival outcome (OS / DSS and competitive risk outcome), LNs, PLNs and LNR between survival group and death group. See Table 1 and S1 Table for details of the OS/DSS group.

## 3.2 Kaplan-Meier marginal regression

We compared OS and DSS through Kaplan-Meier marginal regression and log-rank test based on the counts of LNs, PLNs, and LNR. Due to the short follow-up time, some median survival times cannot be counted temporarily. In OS, the median survival time of PLNs 11–75 groups was 134 months, P<0.001. The median survival time in the group with LNR>54% was 122 months, P < 0.001. In DSS, the median survival time in the group with LNR>54% was 153 months, P < 0.001. Both groups suggest that the more LNs, the more PLNs, and the higher LNR, the prognosis of the group is poor (Fig 2).

**Table 1. Patients characteristics and demographics(OS).**

| Variables | Total (n = 2435) | Survival (n = 1986) | Death (n = 449) | p |
|---|---|---|---|---|
| Age, Median (Q1, Q3) | 54 (42, 65) | 52 (40, 63) | 65 (53, 74) | < 0.001 |
| Sex, n (%) | | | | < 0.001 |
| Female | 1439 (59.1) | 1227 (61.78) | 212 (47.22) | |
| Male | 996 (40.9) | 759 (38.22) | 237 (52.78) | |
| Race, n (%) | | | | 0.084 |
| Black | 185 (7.6) | 151 (7.6) | 34 (7.57) | |
| Others | 173 (7.1) | 152 (7.65) | 21 (4.68) | |
| White | 2077 (85.3) | 1683 (84.74) | 394 (87.75) | |
| Marital, n (%) | | | | < 0.001 |
| Divorced/Separated | 162 (6.65) | 103 (5.19) | 59 (13.14) | |
| Married | 1499 (61.56) | 1217 (61.28) | 282 (62.81) | |
| Single/Unmarried | 462 (18.97) | 405 (20.39) | 57 (12.69) | |
| Widowed/Others | 312 (12.81) | 261 (13.14) | 51 (11.36) | |
| Diagnosis, n (%) | | | | < 0.001 |
| 1998~2000 | 193 (7.93) | 115 (5.79) | 78 (17.37) | |
| 2001~2005 | 499 (20.49) | 336 (16.92) | 163 (36.3) | |
| 2006~2010 | 723 (29.69) | 585 (29.46) | 138 (30.73) | |
| 2011~2016 | 1020 (41.89) | 950 (47.83) | 70 (15.59) | |
| AJCC, n (%) | | | | < 0.001 |
| I~III | 672 (27.6) | 641 (32.28) | 31 (6.9) | |
| IV | 303 (12.44) | 241 (12.13) | 62 (13.81) | |
| Unknown | 1460 (59.96) | 1104 (55.59) | 356 (79.29) | |
| AJCC.T, n (%) | | | | < 0.001 |
| T0~T2 | 719 (29.53) | 678 (34.14) | 41 (9.13) | |
| T3~T4 | 275 (11.29) | 221 (11.13) | 54 (12.03) | |
| Tx/Unknown | 1441 (59.18) | 1087 (54.73) | 354 (78.84) | |
| AJCC.N, n (%) | | | | < 0.001 |
| N0 | 570 (23.41) | 544 (27.39) | 26 (5.79) | |
| N1 | 429 (17.62) | 359 (18.08) | 70 (15.59) | |
| Nx/Unknown | 1436 (58.97) | 1083 (54.53) | 353 (78.62) | |
| AJCC.M, n (%) | | | | < 0.001 |
| NO | 960 (39.43) | 881 (44.36) | 79 (17.59) | |
| Unknown | 1427 (58.6) | 1074 (54.08) | 353 (78.62) | |
| YES | 48 (1.97) | 31 (1.56) | 17 (3.79) | |
| Hist.type, n (%) | | | | < 0.001 |
| MTC | 2117 (86.94) | 1778 (89.53) | 339 (75.5) | |
| MTCAS | 318 (13.06) | 208 (10.47) | 110 (24.5) | |
| Radiotherapy, n (%) | | | | < 0.001 |
| NO/Unknown | 2099 (86.2) | 1777 (89.48) | 322 (71.71) | |
| YES | 336 (13.8) | 209 (10.52) | 127 (28.29) | |
| Radiation.recode, n (%) | | | | < 0.001 |
| EBRT | 229 (9.4) | 124 (6.24) | 105 (23.39) | |
| IRT | 103 (4.23) | 84 (4.23) | 19 (4.23) | |
| Others | 2103 (86.37) | 1778 (89.53) | 325 (72.38) | |
| Chemotherapy, n (%) | | | | < 0.001 |
| NO | 2370 (97.33) | 1953 (98.34) | 417 (92.87) | |
| YES | 65 (2.67) | 33 (1.66) | 32 (7.13) | |

(*Continued*)

**Table 1.** (Continued)

| Variables | Total (n = 2435) | Survival (n = 1986) | Death (n = 449) | p |
|---|---|---|---|---|
| Tumor.size, n (%) | | | | < 0.001 |
| < = 2cm | 910 (37.37) | 830 (41.79) | 80 (17.82) | |
| 2~4cm | 543 (22.3) | 460 (23.16) | 83 (18.49) | |
| Size>4cm | 323 (13.26) | 227 (11.43) | 96 (21.38) | |
| Unknown | 659 (27.06) | 469 (23.62) | 190 (42.32) | |
| Sequence.number2, n (%) | | | | < 0.001 |
| 1 | 2099 (86.2) | 1744 (87.81) | 355 (79.06) | |
| 2~ | 336 (13.8) | 242 (12.19) | 94 (20.94) | |
| Number.1, n (%) | | | | < 0.001 |
| 1 | 1837 (75.44) | 1560 (78.55) | 277 (61.69) | |
| 2~ | 598 (24.56) | 426 (21.45) | 172 (38.31) | |
| Grade, n (%) | | | | < 0.001 |
| I~II | 142 (5.83) | 115 (5.79) | 27 (6.01) | |
| III~IV | 83 (3.41) | 47 (2.37) | 36 (8.02) | |
| Unknown | 2210 (90.76) | 1824 (91.84) | 386 (85.97) | |
| Survival. months, Median (Q1, Q3) | 67 (28, 116) | 71 (31, 122) | 44 (19, 89) | < 0.001 |
| Outcome3, n (%) | | | | < 0.001 |
| 0 | 1986 (81.56) | 1986 (100) | 0 (0) | |
| 1 | 198 (8.13) | 0 (0) | 198 (44.1) | |
| 2 | 251 (10.31) | 0 (0) | 251 (55.9) | |
| Proportion, Median (Q1, Q3) | 0 (0, 0.27) | 0 (0, 0.2) | 0.2 (0, 0.62) | < 0.001 |
| Age.cat, n (%) | | | | < 0.001 |
| ~45 | 750 (30.8) | 688 (34.64) | 62 (13.81) | |
| 46~60 | 824 (33.84) | 704 (35.45) | 120 (26.73) | |
| 61~ | 861 (35.36) | 594 (29.91) | 267 (59.47) | |
| LNR, n (%) | | | | < 0.001 |
| ~25% | 1774 (72.85) | 1532 (77.14) | 242 (53.9) | |
| 25%~53% | 332 (13.63) | 261 (13.14) | 71 (15.81) | |
| 54%~ | 329 (13.51) | 193 (9.72) | 136 (30.29) | |
| PLNs, n (%) | | | | < 0.001 |
| 0 | 1446 (59.38) | 1264 (63.65) | 182 (40.53) | |
| 1~10 | 640 (26.28) | 489 (24.62) | 151 (33.63) | |
| 11~75 | 349 (14.33) | 233 (11.73) | 116 (25.84) | |
| Total_LNs, n (%) | | | | < 0.001 |
| 0~2 | 873 (35.85) | 720 (36.25) | 153 (34.08) | |
| 19~89 | 783 (32.16) | 605 (30.46) | 178 (39.64) | |
| 3~18 | 779 (31.99) | 661 (33.28) | 118 (26.28) | |
| Outcome2, n (%) | | | | < 0.001 |
| 0 | 2237 (91.87) | 1986 (100) | 251 (55.9) | |
| 1 | 198 (8.13) | 0 (0) | 198 (44.1) | |

Abbreviations: EBRT, external beam radiotherapy; IRT, iodine radiotherapy; Sequence.number2, history of previous malignant tumors; Number.1.2~, number of lesions; Outcome0, survival; Outcome1, Disease-specific death; Outcome2, Other deaths.

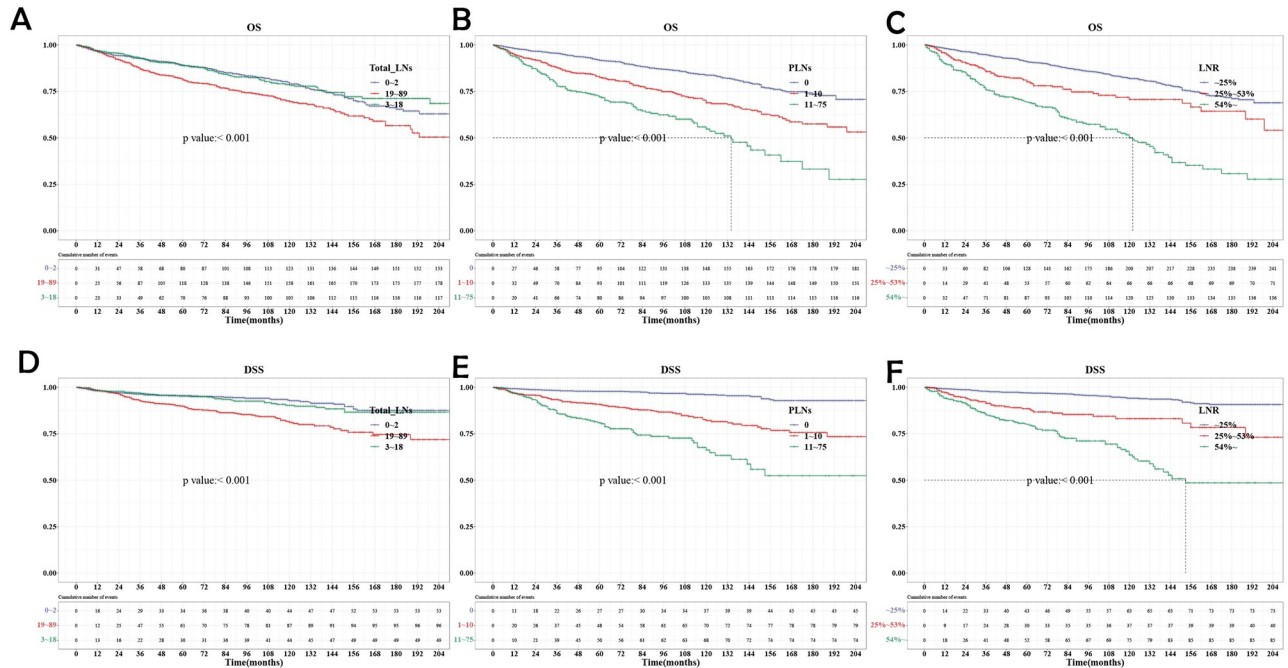

**Fig 2. Kaplan–Meier survival curves.** (A)Kaplan–Meier survival curves of OS stratified by total LNs. (B)Kaplan–Meier survival curves of OS stratified by PLNs. (C)Kaplan–Meier survival curves of OS stratified by LNR. (D)Kaplan–Meier survival curves of DSS stratified by total LNs. (E)Kaplan–Meier survival curves of DSS stratified by PLNs. (F)Kaplan–Meier survival curves of DSS stratified by LNR.

### 3.3 Univariate and multivariate Cox proportional-hazards model

Through univariate and multivariable Cox proportional-hazards models, the hazard ratios values (HR) of all variables for OS of MTC can be seen in Figs 3, S1 and S2. By univariate analysis for OS, age, gender, marital status, radiotherapy, chemotherapy, tumor size, histological grade, AJCC stage, AJCC.T stage, number of lesions, number of total LNs, PLNs, and LNR were independent predictors of MTC. Compared with 0–2, the total LNs removed was 3-18(HR = 0.953, 95%CI[0.749,1.212], P = 0.695), 19–89 (HR = 1.526, 95% CI [1.229, 1.895], P < 0.001). Compared with no metastasis, the number of PLNs was 1–10 (HR = 2.009, 95% CI[1.619, 2.493], P<0.001), 11–25 (HR = 3.557, 95% CI [2.814, 4.497], P<0.001). Compared with less than 25%, the LNR was 25–53% (HR = 1.855, 95% CI[1.424, 2.418], P<0.001), and more than 54% (HR = 3.735, 95% CI[3.026, 4.612], P<0.001).

While in the multivariate Cox proportional-hazards model, compared with 0–2, the total LNs removed were 3-18(HR = 0.889, 95%CI[0.695,1.136], P = 0.347), 19-89(HR = 1.122, 95% CI[0.885,1.423], P = 0.34). Compared with no metastasis, the number of PLNs was 1-10 (HR = 1.535, 95%CI[1.216, 1.939], P<0.001), 11-75(HR = 2.359, 95%CI[1.774,3.137], P<0.001). Compared with less than 25%, the LNR was 25–53%(HR = 1.503, 95%CI [1.128,2.004], P<0.001), and more than 54% (HR = 2.448,95%CI[1.93,3.106], P<0.001). Compared with the results of univariate Cox proportional-hazards models, pathological grade, AJCC stage, AJCC T stage, and the number of total LNs were no longer correlated with prognosis. See Fig 3, S1 and S2 Figs for details.

In DSS, the results showed that age, gender, radiotherapy, chemotherapy, tumor size, multiple lesions, PLNs, and the LNR were independent risk factors for the prognosis of patients with MTC. In the multivariate Cox proportional-hazards models, the number of total LNs was

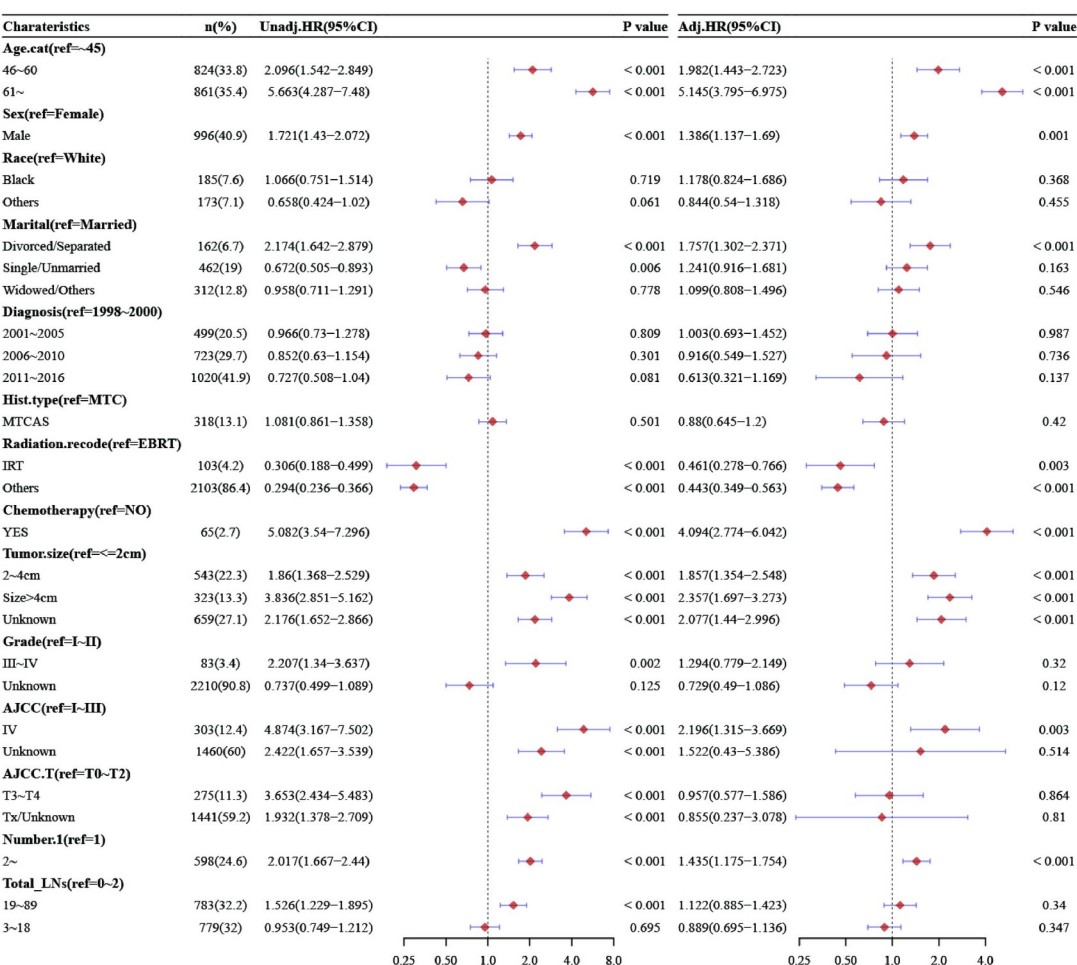

**Fig 3. Univariate and multivariate analysis of OS stratified by total LNs.**

not an independent risk factor, compared with 0–2, the total number of LNs was 3–18 (HR = 1.006, 95% CI [0.675, 1.498], P = 0.978), and the total number of LNs was 19–89 (HR = 1.429, 95% CI [0.989, 2.064], P = 0.057). Compared with no metastasis, the number of PLNs was 1–10 (HR = 2.993, 95% CI [2.037, 4.397], P<0.001), 11–75 (HR = 4.865, 95% CI [3.161, 7.487], P<0.001). Compared with less than 25%, the LNR was 25–53% (HR = 2.38, 95% CI [1.584, 3.578], P<0.001), more than 54% (HR = 4.182, 95%CI [2.949, 5.932], P<0.001). The detailed results can be found in Fig 4, S3 and S4 Figs.

### 3.4 Univariate and multivariate Competing-risks analysis

Through the analysis of the DSS competing-risks model of MTC, the results of the univariate analysis showed that the number of total LNs (19–89), PLNs (1–10,11–75) and positive LNR (25%-53%,>54%), age (46–60,>61), chemotherapy, mode of radiotherapy (others), tumor size (2-4cm,>4cm), number of lesions>1, male gender, diagnosis time (2006–2010,2011–2016), histological grade (III-IV), were poor prognostic factors for MTC. Compared with 0–2, the total number of LNs was 3-18(SHR = 1.134, 95%CI [0.769, 1.674], P = 0.53), and the total number of LNs was 19-89(SHR = 2.294,95%CI[1.639, 3.211], P<0.001), compared with no

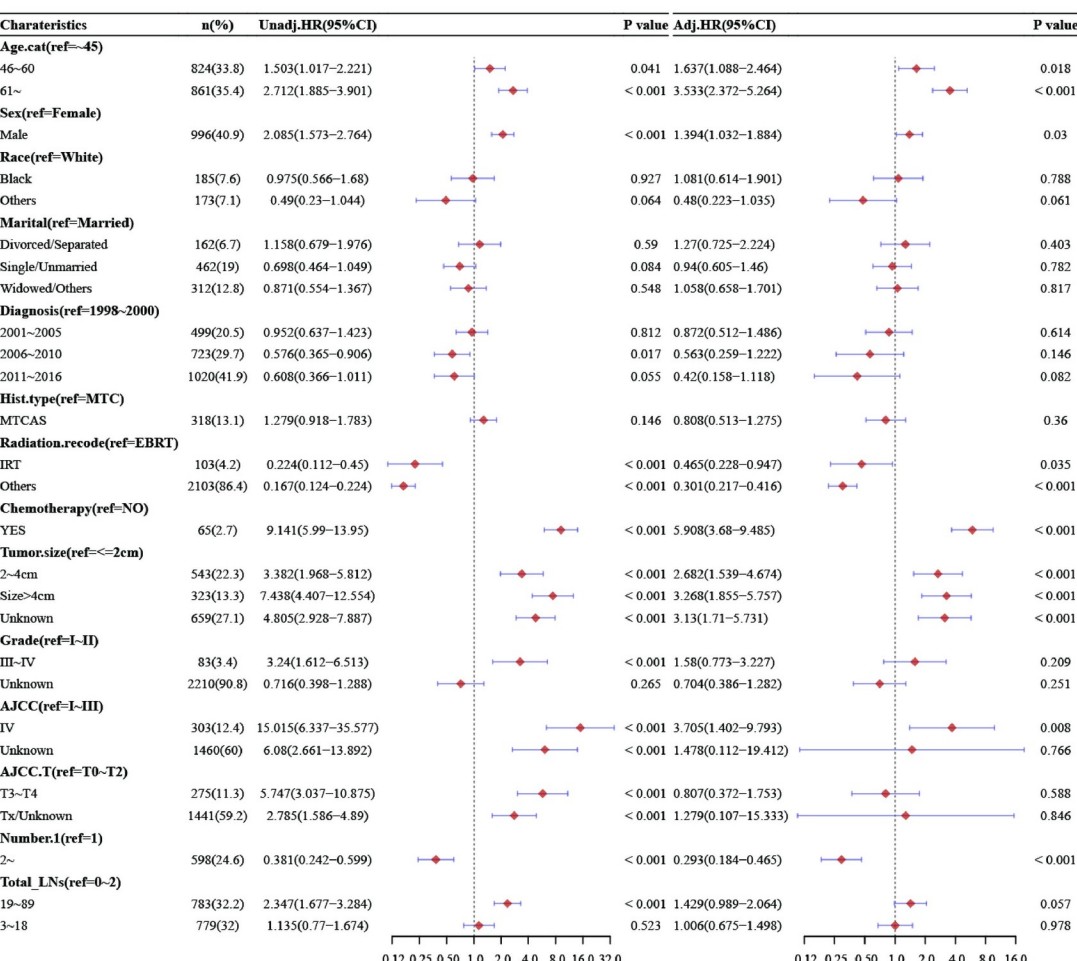

**Fig 4. Univariate and multivariate analysis of DSS stratified by total LNs.**

metastasis, the number of PLNs was 1–10 (SHR = 4.135, 95%CI[2.867, 5.962], P<0.001), 11-75 (SHR = 8.328, 95%CI[5.756, 12.049], P<0.001), compared with less than 25%, the LNR was 25–53%(SHR = 3.335, 95%CI[2.267, 4.905], P<0.001), more than 54%(SHR = 6.979, 95%CI [5.117, 9.519], P<0.001).

While gender, diagnosis time, and histological grade were insignificant in multivariate competing-risks model analysis. Compared with 0–2, the total number of LNs was 3–18 (SHR = 1.01, 95% CI [0.667, 1.531], P = 0.96), and the total number of LNs was 19–89 (SHR = 1.493, 95% CI [1.02, 2.186], P = 0.039), compared with no metastasis, the number of PLNs was 1–10 (SHR = 2.938, 95% CI [1.971, 4.379], P<0.001), 11–75 (SHR = 4.625, 95% CI [2.853, 7.499], P<0.001), compared with less than 25%, the LNR was 25–53% (SHR = 2.253, 95% CI [1.483, 3.421], P<0.001), more than 54% (SHR = 3.493, 95% CI [2.658, 5.85], P<0.001) (Figs 5–7).

### 3.5 The time-dependent AUC value of each variable

The time-dependent AUC (DSS) values of lymph nodes (LNs), positive lymph nodes (PLN), lymph node ratio (LNR), and other clinical factors were compared at 120 months, 156 months,

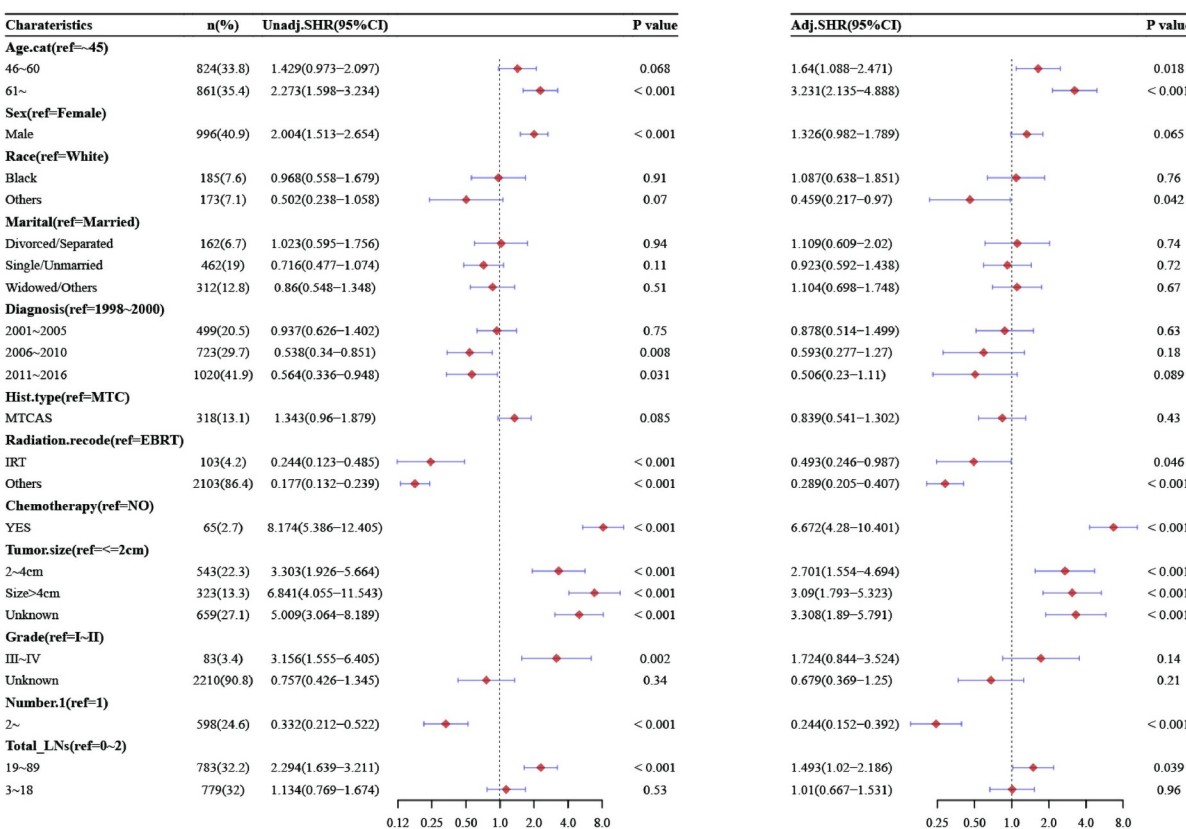

**Fig 5. Multivariate competing-risks analysis in patients with MTC stratified by LNs.**

and 180 months. The analysis showed that LNR and PLNs had a higher significance level than 0.7, indicating their potential as specific diagnostic criteria. Moreover, age, radiotherapy mode, and LNs also demonstrated a significance level greater than 0.6, suggesting their potential in aiding diagnosis. (Table 2).

## 4 Discussion

Medullary thyroid carcinoma has its unique pathological characteristics. However, because of its low incidence rate, the prognosis is mainly limited by the sample size, and there are competitive risk factors in survival statistics. Regional lymph node metastases are present in the majority of patients with palpable MTC. Because these tumors do not take up iodine, lymph node metastases cannot be ablated with radioactive iodine. Surgical clearance is the only effective strategy for eliminating these deposits [6,23].

There is currently significant controversy surrounding the surgical approach to neck lymph node involvement in MTC. Previous recommendations suggested total thyroidectomy with three-compartment lymphadenectomy (central plus bilateral cervicolateral) for patients without evidence of neck lymph node ultrasound involvement, in both primary and completion surgeries [24].

American Thyroid Association (ATA) recommendation in MTC, without lymph node involvement (according to ultrasonographic study) and without systemic metastasis, is prophylactic central (zone 6) lymph node dissection (grade B Recommendation). Biochemical

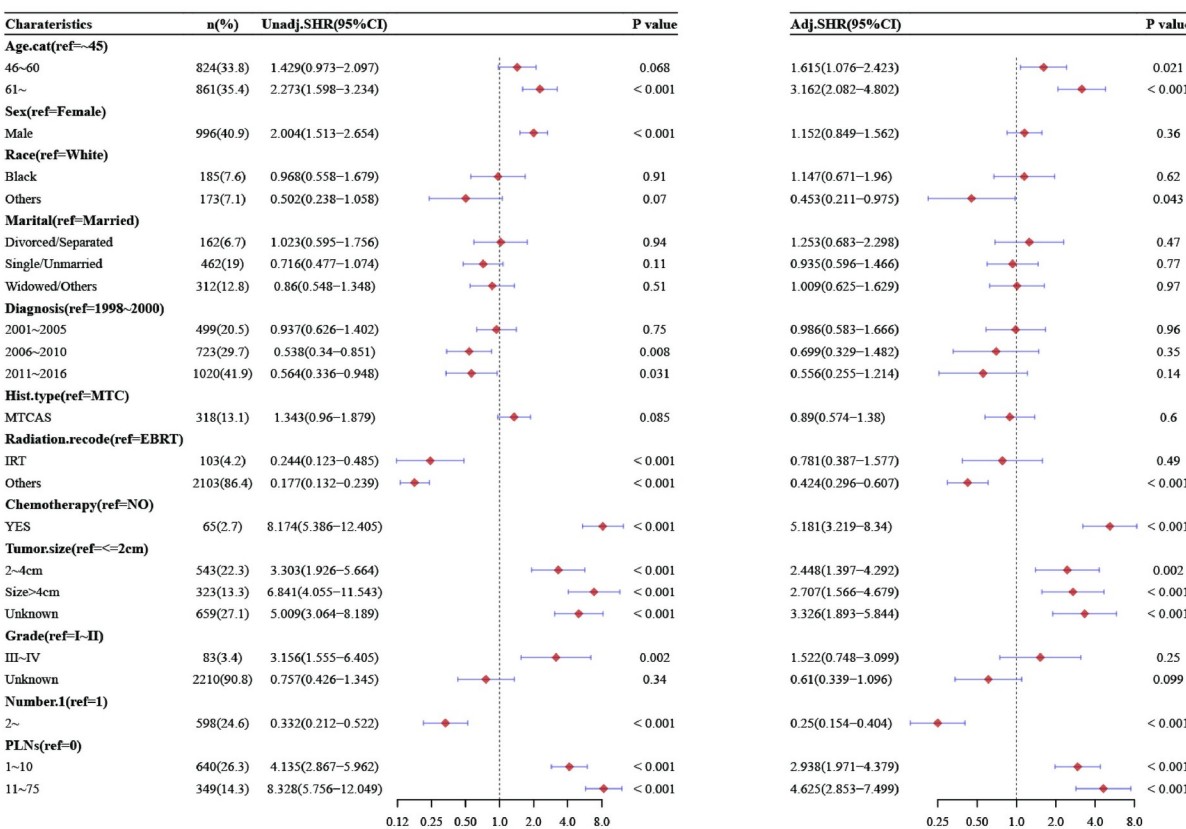

| Charateristics | n(%) | Unadj.SHR(95%CI) | P value | Adj.SHR(95%CI) | P value |
|---|---|---|---|---|---|
| **Age.cat(ref=~45)** | | | | | |
| 46~60 | 824(33.8) | 1.429(0.973–2.097) | 0.068 | 1.615(1.076–2.423) | 0.021 |
| 61~ | 861(35.4) | 2.273(1.598–3.234) | < 0.001 | 3.162(2.082–4.802) | < 0.001 |
| **Sex(ref=Female)** | | | | | |
| Male | 996(40.9) | 2.004(1.513–2.654) | < 0.001 | 1.152(0.849–1.562) | 0.36 |
| **Race(ref=White)** | | | | | |
| Black | 185(7.6) | 0.968(0.558–1.679) | 0.91 | 1.147(0.671–1.96) | 0.62 |
| Others | 173(7.1) | 0.502(0.238–1.058) | 0.07 | 0.453(0.211–0.975) | 0.043 |
| **Marital(ref=Married)** | | | | | |
| Divorced/Separated | 162(6.7) | 1.023(0.595–1.756) | 0.94 | 1.253(0.683–2.298) | 0.47 |
| Single/Unmarried | 462(19) | 0.716(0.477–1.074) | 0.11 | 0.935(0.596–1.466) | 0.77 |
| Widowed/Others | 312(12.8) | 0.86(0.548–1.348) | 0.51 | 1.009(0.625–1.629) | 0.97 |
| **Diagnosis(ref=1998~2000)** | | | | | |
| 2001~2005 | 499(20.5) | 0.937(0.626–1.402) | 0.75 | 0.986(0.583–1.666) | 0.96 |
| 2006~2010 | 723(29.7) | 0.538(0.34–0.851) | 0.008 | 0.699(0.329–1.482) | 0.35 |
| 2011~2016 | 1020(41.9) | 0.564(0.336–0.948) | 0.031 | 0.556(0.255–1.214) | 0.14 |
| **Hist.type(ref=MTC)** | | | | | |
| MTCAS | 318(13.1) | 1.343(0.96–1.879) | 0.085 | 0.89(0.574–1.38) | 0.6 |
| **Radiation.recode(ref=EBRT)** | | | | | |
| IRT | 103(4.2) | 0.244(0.123–0.485) | < 0.001 | 0.781(0.387–1.577) | 0.49 |
| Others | 2103(86.4) | 0.177(0.132–0.239) | < 0.001 | 0.424(0.296–0.607) | < 0.001 |
| **Chemotherapy(ref=NO)** | | | | | |
| YES | 65(2.7) | 8.174(5.386–12.405) | < 0.001 | 5.181(3.219–8.34) | < 0.001 |
| **Tumor.size(ref=<=2cm)** | | | | | |
| 2~4cm | 543(22.3) | 3.303(1.926–5.664) | < 0.001 | 2.448(1.397–4.292) | 0.002 |
| Size>4cm | 323(13.3) | 6.841(4.055–11.543) | < 0.001 | 2.707(1.566–4.679) | < 0.001 |
| Unknown | 659(27.1) | 5.009(3.064–8.189) | < 0.001 | 3.326(1.893–5.844) | < 0.001 |
| **Grade(ref=I~II)** | | | | | |
| III~IV | 83(3.4) | 3.156(1.555–6.405) | 0.002 | 1.522(0.748–3.099) | 0.25 |
| Unknown | 2210(90.8) | 0.757(0.426–1.345) | 0.34 | 0.61(0.339–1.096) | 0.099 |
| **Number.1(ref=1)** | | | | | |
| 2~ | 598(24.6) | 0.332(0.212–0.522) | < 0.001 | 0.25(0.154–0.404) | < 0.001 |
| **PLNs(ref=0)** | | | | | |
| 1~10 | 640(26.3) | 4.135(2.867–5.962) | < 0.001 | 2.938(1.971–4.379) | < 0.001 |
| 11~75 | 349(14.3) | 8.328(5.756–12.049) | < 0.001 | 4.625(2.853–7.499) | < 0.001 |

**Fig 6. Multivariate competing-risks analysis in patients with MTC stratified by PLNs.**

results can help determine the extent of lymphatic dissection. If the level of calcitonin is higher than 20 pg/ml, a prophylactic ipsilateral central and ipsilateral lateral dissection is recommended and if it is higher than 200 pg/ml, a prophylactic dissection in uninvolved contralateral lateral neck compartments is recommended. All guidelines and review articles recommend central and lateral dissection if lymphadenopathy is confirmed in preoperative examinations. In patents with locally advanced or metastatic MTC, in addition to thyroidectomy, dissection of compartments with involved lymph nodes is often recommended. For this reason, during dissection of the central and lateral zones of the neck, proceedings that cause damage to speech, swallowing, shoulder movements and parathyroid glands should be avoided [2,25].

The impact of lymph node status on the prognosis and staging of medullary thyroid cancer in previous studies requires further consolidation. In the ATA guidelines for the management of medullary thyroid cancer, it was pointed out that quantitative assessment of lymph node metastases, 1–10 (N1), 11–20 (N2), and more than 20 (N3), is an important prognostic classifier that should be incorporated into the AJCC staging systems, which currently includes N1a and N1b categories referring only to qualitative involvement of lymph node compartments [2].

This study aimed to analyze the prognostic value of total number, positive number, and positive proportion of LNs in patients with MTC who underwent total thyroidectomy and neck dissection using the SEER database and competing risk models. Through a large sample analysis, this study provides evidence for the prognostic value of lymph nodes in MTC

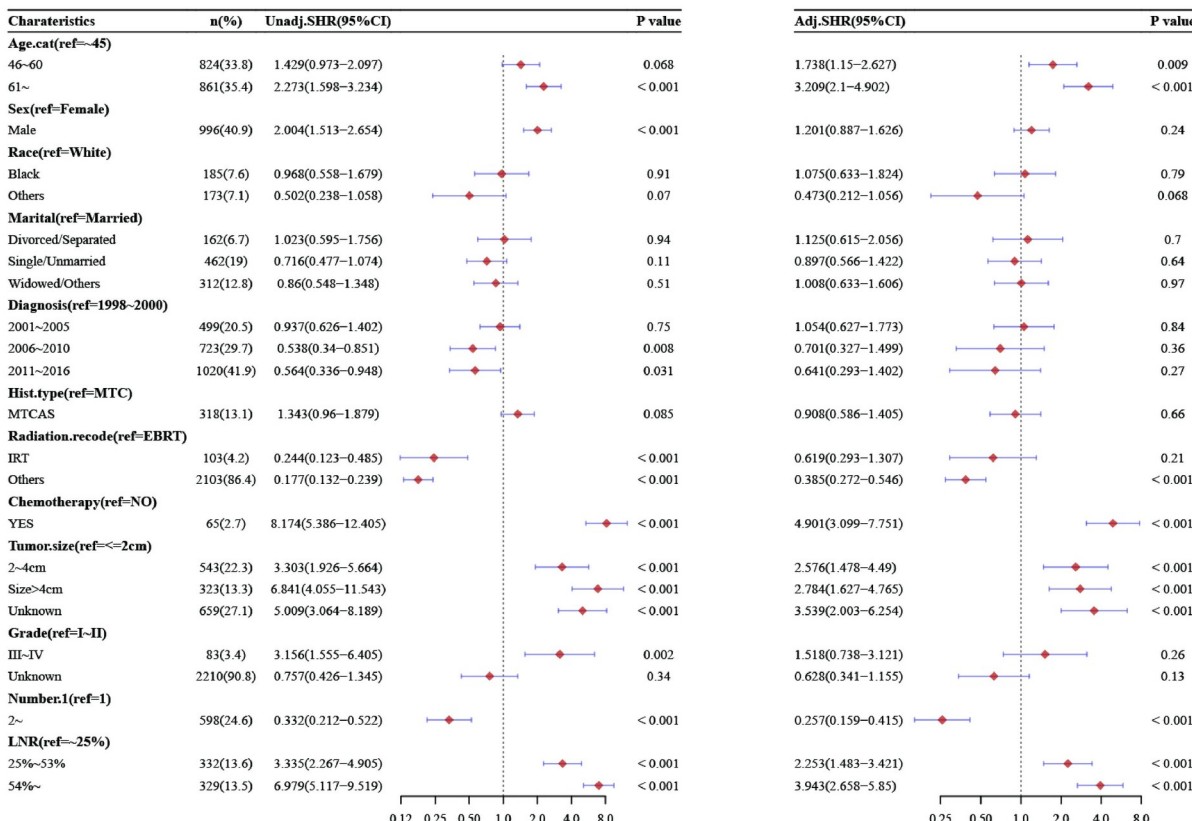

**Fig 7. Multivariate competing-risks analysis in patients with MTC stratified by LNR.**

**Table 2. Time-dependent AUC values (DSS).**

|  | t = 120 | t = 156 | t = 180 |
|---|---|---|---|
| Age.cat | 0.656 | 0.62 | 0.64 |
| Sex | 0.611 | 0.614 | 0.581 |
| Race | 0.511 | 0.52 | 0.521 |
| Marital | 0.515 | 0.523 | 0.53 |
| Diagnosis | 0.365 | 0.265 | 0.197 |
| Hist. type | 0.432 | 0.366 | 0.295 |
| Radiation.recode | 0.652 | 0.635 | 0.624 |
| Chemotherapy | 0.549 | 0.54 | 0.548 |
| Tumor. size | 0.567 | 0.473 | 0.475 |
| Grade | 0.554 | 0.538 | 0.531 |
| AJCC | 0.569 | 0.55 | 0.546 |
| AJCC.T | 0.528 | 0.521 | 0.519 |
| AJCC.N | 0.572 | 0.553 | 0.549 |
| AJCC.M | 0.454 | 0.466 | 0.469 |
| Number.1 | 0.545 | 0.549 | 0.549 |
| LNR | 0.713 | 0.737 | 0.714 |
| PLNs | 0.757 | 0.756 | 0.743 |
| Total_LNs | 0.642 | 0.627 | 0.624 |

patients, guiding clinical diagnosis, treatment, and surgical decision-making. In previous studies on MTC, the prognosis is closely related to various factors, including gender, age at diagnosis, local tumor invasion, LN metastasis, distant metastasis (DM), and response to initial treatment [2]. In our study's multivariate competing-risks model analysis, LNs, PLNs, LNR, age, race, mode of radiotherapy (others), tumor size, and the number of lesions are independent risk factors for MTC.

In the multivariate competing-risks model analysis, compared with 0–2, the total number of LNs was 19–89 (SHR = 1.493, 95% CI [1.02, 2.186], P = 0.039), which was an independent risk factor for prognosis. In contrast, the total number of LNs examined for MTC was not in the multivariate Cox regression, indicating that the total number of LNs was still an independent risk factor for prognosis after considering the competitive risk. This result should be the bias caused by competitive risk events, which shows the same result as the Kaplan-Meier marginal regression. For patients with medullary thyroid carcinoma, total thyroidectomy combined with central compartment lymph node dissection is recommended by the American Association of endocrine surgeons and most guidelines, regardless of the status of lymph node involvement [2,26]. In our study, the total number of LNs was 0–2, which means the LNs were not or not thoroughly cleaned. As an independent risk factor, it also reflected the necessity of lymph node dissection in patients with medullary thyroid carcinoma. In breast and lung cancer, studies have also reported the value of negative LNs as a prognostic factor [27,28]. The possible reason is that the higher number of LNs detected reflects the higher surgical quality of lymph node dissection, which also means the higher detection rate of PLNs and is related to the body's immune response to the tumor.

In the multivariate competing-risks model, compared with no metastasis, the number of PLNs was 1–10 (SHR = 2.938, 95% CI [1.971, 4.379], P<0.001) and 11–75 PLNs (SHR = 4.625, 95% CI [2.853, 7.499], P<0.001). While in the multivariate Cox proportional-hazards model of DSS, compared with no metastasis, the number of PLNs was 1–10 (HR = 2.993, 95% CI [2.037, 4.397], P<0.001) and 11–75 PLNs (HR = 4.865, 95% CI [3.161, 7.487], P<0.001). This conclusion suggests that the results of the competitive risk model are the same as that of the Cox proportional-hazards model, indicating that competitive risks do not affect the prognosis of PLNs. Currently, the n category of MTC by the American Joint Commission on Cancer (AJCC) is defined by the location of PLNs. Some studies have shown that the number of PLNs may be more related to the risk of death in MTC patients, which is consistent with our analysis results [8,29].

Some studies have shown that LNR is a better independent predictor of MTC because the number of LNs and PLNs may be affected by pathological identification and surgical techniques, but LNR can reduce the impact of surgical procedures [9]. Although some studies have shown that LNR does not correlate with OS and DSS [30], in our multivariate competing-risks model, compared with less than 25%, 25%-53% (SHR = 2.253, 95% CI [1.483, 3.421], P<0.001) and more than 54% (SHR = 3.943, 95% CI [2.2658, 5.85], P<0.001). In the multivariate Cox proportional-hazards model of DSS, compared with less than 25%, 25%-53% (HR = 2.38, 95% CI [1.584, 3.578], P<0.001) and more than 54% (HR = 4.182, 95% CI [2.949, 5.932], P<0.001). It shows that the LNR is determined as an independent risk factor. Still, considering the competitive risk, we found that the Cox proportional-hazards model overestimated the impact of the LNR on the prognosis. In addition, because the prediction ability of LNR is limited when all LNs are positive or negative, some studies have shown that the log odd of positive lymph nodes (LODDS) can more accurately reflect the prognostic value, but further statistical research of large samples is still needed [31,32].

Although the Cox proportional-hazards model results show that male relative to female, divorced / single relative to married, external radiation radiotherapy relative to radioactive

iodine therapy has a worse prognosis, while some studies have also reached similar conclusions, this effect is not found in the competing-risks model. We infer that these statistical results are due to the bias caused by competing risk events [33,34].

This study has the following advantages. Firstly, the sample size was large, and the follow-up time was long. By searching the SEER database, 2435 patients were included in the statistics, and the median follow-up time was 67 months. Second, there are many variables in the study. In addition to the primary research variables (the total LNs, PLNs, and positive LNR), this study includes multiple variables such as age, radiotherapy, chemotherapy, and AJCC stage. Thirdly, this study excludes the impact of competitive events through the competing-risks model, which can more accurately evaluate the relationship between predictive variables and outcome events.

Although we have systematically studied the prognostic analysis of MTC, there are also many limitations. Firstly, potential selection bias and possible residual confusion in the SEER database cannot be avoided, and most of the data in the SEER database are white race, so there was potential racial heterogeneity that could not be extrapolated to other human species. Moreover, there is insufficient information on many variables, such as no detailed treatment plan and drug dose. MTC is also closely related to factors such as serum calcitonin level, and we also lack this part of the data [35]. Thirdly, the limitations inherent in retrospective studies are inevitable.

## 5 Conclusion

In conclusion, our study utilized the SEER database to establish a competing-risks model for patients with medullary thyroid carcinoma who underwent total thyroidectomy, and we demonstrated the prognostic significance of the total number of LNs, PLNs, and positive LNR. Specifically, we found that the number and proportion of PLNs have a definite prognostic significance, and the total number of LNs becomes an independent risk factor for prognosis after excluding the competitive risk. These findings can help clinicians more accurately evaluate the prognosis of patients with medullary thyroid carcinoma and provide important insights for clinical treatment.

## Supporting information

**S1 Table. Patients characteristics and demographics(DSS).**
(DOCX)

**S1 Fig. Univariate and multivariate analysis of OS stratified by PLNs.**
(TIF)

**S2 Fig. Univariate and multivariate analysis of OS stratified by total LNR.**
(TIF)

**S3 Fig. Univariate and multivariate analysis of DSS stratified by PLNs.**
(TIF)

**S4 Fig. Univariate and multivariate analysis of DSS stratified by LNR.**
(TIF)

## Author Contributions

**Conceptualization:** Xin Ren.

**Data curation:** Yanlin Li.

**Formal analysis:** Lei Cai.

**Supervision:** Jian Wen.

**Validation:** Jian Wen.

**Visualization:** Yujia Sun.

**Writing – original draft:** Fangjian Shang, Xiaodan Liu.

**Writing – review & editing:** Fangjian Shang, Xiaodan Zhai.

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
