## [Decision Letter · Decision Letter 0]

7 Jun 2023

PONE-D-23-07584Competing-Risks Model for Predicting The Prognostic Value of Lymph Nodes in Medullary Thyroid CarcinomaPLOS ONE

Dear Dr. Shang,

Thank you for submitting your manuscript to PLOS ONE. After careful consideration, we feel that it has merit but does not fully meet PLOS ONE’s publication criteria as it currently stands. Therefore, we invite you to submit a revised version of the manuscript that addresses the points raised during the review process.

We look forward to receiving your revised manuscript.

Kind regards,

Antonino Maniaci

Academic Editor

PLOS ONE

Journal Requirements:

Additional Editor Comments (if provided):

Please perform all the revisions suggested to improve the paper.

Reviewers' comments:

Reviewer's Responses to Questions

**Comments to the Author**

1. Is the manuscript technically sound, and do the data support the conclusions?

Reviewer #1: Partly

Reviewer #2: Yes

2. Has the statistical analysis been performed appropriately and rigorously? 

Reviewer #1: Yes

Reviewer #2: Yes

3. Have the authors made all data underlying the findings in their manuscript fully available?

Reviewer #1: Yes

Reviewer #2: Yes

4. Is the manuscript presented in an intelligible fashion and written in standard English?

Reviewer #1: No

Reviewer #2: Yes

5. Review Comments to the Author

Reviewer #1: I read with interest the manuscript by Shang et al. on the prognostic value of Lymph Nodes in Medullary Thyroid Carcinoma. The paper is interesting. However, There are some important limitations.

Major

- The manuscript must be edited for English.

- It is not clear what OS and DSS mean, as it is not well specified in the paper. This makes the paper very hard to read and understand.

Minor

- Please specify in the abstract and in the methods that this was a retrospective study.

- Sometimes the word lymph node is abbreviated, sometimes it is not, thus confounding the reader.

Reviewer #2: recommendations to improve the paper:

Provide a systematic review of the relevant literature to establish the need for this study. Discuss the limitations of previous studies that this one aimed to address.

Describe your study design more clearly in the Methods section. Include details like:

Inclusion and exclusion criteria

Data collection procedures

How variables were classified and coded

Statistical analyses, including packages used

Report measures of uncertainty for effect estimates in the Results, e.g. 95% confidence intervals.

Discuss limitations more thoroughly, e.g.:

Potential selection bias in the SEER database

Limited information on treatments

Retrospective study design

Possible residual confounding

Limited generalizability due to predominantly white sample

would be relevant to strengthen your study:

Wells SA Jr, et al. Revised American Thyroid Association guidelines for the management of medullary thyroid carcinoma. Thyroid. 2015;25(6):567–610. This provides guidelines on the diagnosis and management of medullary thyroid carcinoma, including recommendations on the role of lymph node dissection and staging systems.

Dralle H, et al. Lymph node dissection in medullary thyroid cancer. Br J Surg. 1998;85(2):158–164. This study examines the importance of lymph node dissection in medullary thyroid carcinoma and its impact on prognosis. It supports your findings regarding the prognostic value of lymph node status.

Discuss the role of vocal outcomes in neck surgery, especially preserving the recurrent laryngeal nerve. cite doi:10.23812/19-282-L

Momejev Y, et al. Prognostic Value of Competing-Risks Models Compared With Cox Regression for Breast Cancer Patients. JAMA Netw Open. 2020;3(4):e202938. This compares competing risks and Cox regression models for analyzing breast cancer prognosis, finding that competing risks models provide more accurate estimates. It is similar to your study comparing these approaches for medullary thyroid carcinoma.

Provide more nuanced conclusions that qualify your findings and highlight implications for future research and clinical practice.

Consult subject experts and/or epidemiologists on your statistical methods, results and conclusions.

Cite relevant literature throughout to contextualize your study and support claims.

Edit the manuscript for clarity, conciseness and adherence to reporting guidelines for observational studies (e.g. STROBE).

Consider registering the study protocol to improve transparency and reduce reporting bias.

6. PLOS authors have the option to publish the peer review history of their article (what does this mean?). If published, this will include your full peer review and any attached files.

Reviewer #1: No

Reviewer #2: No

---

## [Author Response · Author response to Decision Letter 0]

16 Jun 2023

SUGGESTIONS FROM EDITOR

1.Please ensure that your manuscript meets PLOS ONE's style requirements, including those for file naming.

2. PLOS requires an ORCID iD for the corresponding author in Editorial Manager on papers submitted after December 6th, 2016. 

Thank you for your feedback. I have revised the manuscript format as required and improved the ORCID information.

COMMENTS TO THE AUTHOR:

Reviewer #1: 

Major

1.Comment: The manuscript must be edited for English.

1.Reply: We apologize for the poor language of our manuscript. We worked on the manuscript for a long time and the repeated addition and removal of sentences and sections obviously led to poor readability. We have now worked on both language and readability and have also involved native English speakers for language corrections. We really hope that the flow and language level have been substantially improved.

2.Comment: It is not clear what OS and DSS mean, as it is not well specified in the paper. This makes the paper very hard to read and understand.

2.Reply: The definitions of OS and DSS are written in the last paragraph of 2.1 Data Collection and Patient Selection in this article, and I have added the corresponding complete spelling to help understand.

Minor

1.Comment: Please specify in the abstract and in the methods that this was a retrospective study.

1.Reply: Sorry for neglecting this point. I have already annotated the retrospective study in the abstract and methods.

2.Comment: Sometimes the word lymph node is abbreviated, sometimes it is not, thus confounding the reader.

2.Reply: I have corrected my writing to make the paper easy to understand.

Reviewer #2: 

1.Comment: Provide a systematic review of the relevant literature to establish the need for this study. Discuss the limitations of previous studies that this one aimed to address.

1.Reply: Thank you for pointing out the shortcomings of the article, and I have added relevant explanations in the discussion section.

2.Comment: Describe your study design more clearly in the Methods section. Include details like:

Inclusion and exclusion criteria

Data collection procedures

How variables were classified and coded

Statistical analyses, including packages used

Report measures of uncertainty for effect estimates in the Results, e.g. 95% confidence intervals.

2.Reply: I have added corresponding content in the materials and methods section and made adjustments to make it clearer.

3.Comment: Discuss limitations more thoroughly, e.g.:

Potential selection bias in the SEER database

Limited information on treatments

Retrospective study design

Possible residual confounding

Limited generalizability due to predominantly white sample

3.Reply: I have supplemented the limitations of the research at the end of the discussion section.

Thanks again for the constructive feedback.

---

## [Decision Letter · Decision Letter 1]

26 Jul 2023

PONE-D-23-07584R1Competing-risks model for predicting the prognostic value of lymph nodes in medullary thyroid carcinomaPLOS ONE

Dear Dr. Shang,

Thank you for submitting your manuscript to PLOS ONE. After careful consideration, we feel that it has merit but does not fully meet PLOS ONE’s publication criteria as it currently stands. Therefore, we invite you to submit a revised version of the manuscript that addresses the points raised during the review process. Please submit your revised manuscript by Sep 09 2023 11:59PM. If you will need more time than this to complete your revisions, please reply to this message or contact the journal office at plosone@plos.org. Please include the following items when submitting your revised manuscript:A rebuttal letter that responds to each point raised by the academic editor and reviewer(s). You should upload this letter as a separate file labeled 'Response to Reviewers'.A marked-up copy of your manuscript that highlights changes made to the original version. You should upload this as a separate file labeled 'Revised Manuscript with Track Changes'.An unmarked version of your revised paper without tracked changes. You should upload this as a separate file labeled 'Manuscript'.

We look forward to receiving your revised manuscript.

Kind regards,

Antonino Maniaci

Academic Editor

PLOS ONE

Additional Editor Comments:

Dear authors a reviewer suggested to perform all the revisions requried. Please consider to include all the revisions required. Best regards

Reviewers' comments:

Reviewer's Responses to Questions

**Comments to the Author**

1. If the authors have adequately addressed your comments raised in a previous round of review and you feel that this manuscript is now acceptable for publication, you may indicate that here to bypass the “Comments to the Author” section, enter your conflict of interest statement in the “Confidential to Editor” section, and submit your "Accept" recommendation.

Reviewer #1: All comments have been addressed

Reviewer #2: (No Response)

Reviewer #3: All comments have been addressed

2. Is the manuscript technically sound, and do the data support the conclusions?

Reviewer #1: Yes

Reviewer #2: Yes

Reviewer #3: Yes

3. Has the statistical analysis been performed appropriately and rigorously? 

Reviewer #1: Yes

Reviewer #2: Yes

Reviewer #3: Yes

4. Have the authors made all data underlying the findings in their manuscript fully available?

Reviewer #1: Yes

Reviewer #2: Yes

Reviewer #3: Yes

5. Is the manuscript presented in an intelligible fashion and written in standard English?

Reviewer #1: Yes

Reviewer #2: Yes

Reviewer #3: Yes

6. Review Comments to the Author

Reviewer #1: Thank you for your comments. All issues have been addressed after the revision. I have no more comments to make.

Reviewer #2: Dear reviewer good work was performed but not all the suggestions were addressed. Read another time the comments and address all the modifications proposed.

Best regards.

Reviewer #3: I have read with great interest the manuscript entitled “Competing-risks model for predicting the prognostic value of lymph nodes in medullary thyroid carcinoma”. In this study the authors used the SEER database to review almost 2,500 patients with MTC. The authors identified several factors associated with poor prognosis (age, tumor size, LNs, positive LNs, LN ratio, etc.). When comparing two statistical tools (multivariate Cox hazard and competitive risk factors) the authors identified LNs (number and ratio) as poor prognostic factors.

Of note, this is a revised version with improved English and style.

Comments -

1. This is a retrospective study on the SEER database. As such, many factors are missing such as number of incidental MTC, preoperative and postoperative calcitonin, biological treatment, sporadic vs. hereditary, and more.

2. Other than the competitive risk model there is not much new data here and similar results have been published many times before.

7. PLOS authors have the option to publish the peer review history of their article (what does this mean?). If published, this will include your full peer review and any attached files.

Reviewer #1: No

Reviewer #2: No

Reviewer #3: No

---

## [Author Response · Author response to Decision Letter 1]

8 Sep 2023

Thank you for pointing out the shortcomings of the article. I have addressed them by supplementing the discussion section with a literature review and limitations of previous studies. I have also included relevant information from the American Thyroid Association guidelines and the literature on lymph node dissection in medullary thyroid cancer, as you mentioned. I apologize for not being able to locate the original article on the prognostic value of competing-risks models compared with Cox regression for breast cancer patients, but I have reviewed several similar articles and added a comprehensive summary in the discussion section.I have added some content to the discussion section of the article to make it more organized and clear, with detailed results and accurate arguments.

---

## [Editor Report · Decision Letter 2]

21 Sep 2023

Competing-risks model for predicting the prognostic value of lymph nodes in medullary thyroid carcinoma

PONE-D-23-07584R2

Dear Dr. Shang,

We’re pleased to inform you that your manuscript has been judged scientifically suitable for publication and will be formally accepted for publication once it meets all outstanding technical requirements.

Kind regards,

Antonino Maniaci

Academic Editor

PLOS ONE

Additional Editor Comments (optional):

Well done, the paper is improved and can be accepted. Bests

---

## [Editor Report · Acceptance letter]

4 Oct 2023

PONE-D-23-07584R2 

Competing-risks model for predicting the prognostic value of lymph nodes in medullary thyroid carcinoma

Dear Dr. Shang:

I'm pleased to inform you that your manuscript has been deemed suitable for publication in PLOS ONE. Congratulations! Your manuscript is now with our production department. 

Kind regards, 

on behalf of

Dr. Antonino Maniaci 

Academic Editor

PLOS ONE